# Dopant-Free Triazatruxene-Based Hole Transporting Materials with Three Different End-Capped Acceptor Units for Perovskite Solar Cells

**DOI:** 10.3390/nano10050936

**Published:** 2020-05-13

**Authors:** Da Rim Kil, Chunyuan Lu, Jung-Min Ji, Chul Hoon Kim, Hwan Kyu Kim

**Affiliations:** 1Global GET-Future Lab. Korea University, 2511 Sejong-ro, Sejong 339-700, Korea; ekfla6024@korea.ac.kr (D.R.K.); lcy99168@gmail.com (C.L.); manbbong@korea.ac.kr (J.-M.J.); 2Department of Advanced Materials Chemistry, Korea University, 2511 Sejong-ro, Sejong 339-700, Korea; chulhoon@korea.ac.kr

**Keywords:** perovskite solar cell, dopant-free hole transporting materials, triazatruxene-based organic compound, end-capped acceptor unit, hole mobility

## Abstract

A series of dopant-free D-π-A structural hole-transporting materials (HTMs), named as SGT-460, SGT-461, and SGT-462, incorporating a planar-type triazatruxene (TAT) core, thieno[3,2-*b*]indole (TI) π-bridge and three different acceptors, 3-ethylthiazolidine-2,4-dione (ED), 3-(dicyano methylidene)indan-1-one (DI), and malononitrile (MN), were designed and synthesized for application in perovskite solar cells (PrSCs). The effect of three acceptor units in star-shaped D-π-A structured dopant-free HTMs on the photophysical and electrochemical properties and the photovoltaic performance were investigated compared to the reference HTM of 2,2′,7,7′-tetrakis[*N*,*N*-di(4-methoxyphenyl)amino]-9,9′-spirobifluorene (spiro-OMeTAD). Their highest occupied molecular orbital (HOMO) energy levels were positioned for efficient hole extraction from a MAPbCl_3−_*_x_*I*_x_* layer (5.43 eV). The hole mobility values of the HTMs without dopants were determined to be 7.59 × 10^−5^ cm^2^ V^−1^ s^−1^, 5.13 × 10^−4^ cm^2^ V^−1^ s^−1^, and 7.61 × 10^−4^ cm^2^ V^−1^ s^−1^ for SGT-460-, SGT-461-, and SGT-462-based films. The glass transition temperature of all HTMs showed higher than that of the spiro-OMeTAD. As a result, the molecular engineering of a planar donor core, π-bridge, and end-capped acceptor led to good hole mobility, yielding 11.76% efficiency from SGT-462-based PrSCs, and it provides a useful insight into the synthesis of the next-generation of HTMs for PrSC application.

## 1. Introduction

The organic–inorganic hybrid halide perovskite solar cells (PrSCs) with methyl ammonium lead iodide have attracted great attention as an efficient conversion technology and as a most promising approach due to their low-cost production process, simple device architecture, and excellent photovoltaic performance [1,2,3,4,5,6]. Recently, PrSCs have achieved 25.2% of the power conversion efficiency (PCE) [7], showing that PrSCs are the most viable alternatives to silicon-based solar cells. In particular, the unique properties of halide organic-inorganic perovskite (HOIP), such as broad light absorption range, high extinction coefficient, low exciton binding energy, long carrier diffusion, and charge carrier mobility, make them the most promising thin-film solar cells [8,9]. To support and optimize the excellent properties of HOIP, many researchers have devoted their efforts to studying the other components of PrSCs, which are electron transporting materials, hole transporting materials (HTMs), and counter electrodes [4]. Among them, much attention has been paid to HTMs because they play a crucial role in PrSCs by extracting the photo-generated holes of the HOIP layer, preventing backflow of electrons from the HIOP layer to the counter electrode, and protecting the moisture-sensitive HOIP layer [10,11]. Moreover, they also determine the open circuit photovoltage (*V*_OC_) of PrSCs [12]. Due to their excellent performance, 2,2′,7,7′-tetrakis[*N*,*N*-di(4-methoxyphenyl)amino]-9,9′-spirobifluorene as an organic small molecule HTM has become the most successful and popular HTMs in PrSCs [11]. However, their drawbacks, such as synthetic complexity, expensive cost, moderate glass transition temperature, and low hole mobility, make the commercialization of PrSCs difficult [13,14]. Furthermore, the use of dopants to achieve competitive efficiencies damages the long-term stability of a PrSCs [8,15]. Therefore, developing cheap and efficient dopant-free HTMs is one of the major priorities [16,17,18]. 

Most of efficient dopant-free HTMs have the donor−π−acceptor (D-π-A) architecture because they allow the low band gap, ionic character enhancement by avoiding the use of dopants, and strong dipolar intermolecular interaction [14]. Based on the D-π-A architecture, many researchers have approached various strategies to solve the above disadvantages [15,19]. One of the strategies is the modification of molecular geometry, which can affect its dipole moment, solubility, and film quality [15]. The linear [20,21,22,23], star-shaped [24], and butterfly-shaped geometry are representative [25,26]. The other is the modification of chemical moieties such as core unit, electron donor, π-bridge, and acceptor. The dopant-free HTMs based on various moieties, such as diphenylamine [27], triphenylamine [18,28], carbazole as electron donors [29], thiophene [30], 3,4-ethylenedioxythiophene [31], naphtho-dithiophene [32], spiro[fluorene-9,9′-xanthene] [33], and triazatruxene (TAT) as core units, electron donors, and π-bridges [24], diketopyrrolopyrrole [34], isoindigo [35], and benzothiadiazole as electron acceptors [36,37], have been applied in PrSCs. 

In this work, as shown in Figure 1, three new dopant-free HTMs, coded as SGT-460, SGT-461, and SGT-462, were designed and synthesized. TAT as the central core donor, three thieno[3,2-*b*]indole (TI) moieties as π-bridges, and three different acceptors, 3-ethylthiazolidine-2,4-dione (ED), 3-(dicyano methylidene)indan-1-one (DI), and malononitrile (MN), as the acceptors and the end-capped groups, were used to construct the star-shaped SGT-460, SGT-461, and SGT-462, respectively.

The TAT moiety is the most commonly used moiety for HTMs due to its easy modification of its photophysical and electrochemical properties, strong π-π stacking ability, and excellent charge transporting ability as well as the stabilization ability of the oxidized radical cation form during operation by the nitrogen atom of indole moiety [38]. On the other hand, TI-based moieties were employed as π-bridges in an organic sensitizer for dye-sensitized solar cells (DSSCs), which exhibited a strong electron-donating ability and co-planarity [39]. To increase the π-conjugation length and adjust the energy level, TI moieties were introduced. In addition, we assumed that the presence of alkyl groups in the TI moiety could increase the solubility for organic solvents [39]. Three different acceptors were introduced to investigate the effects of the TAT core-based HTMs on molecular and photovoltaic properties. 

## 2. Results and Discussion

### 2.1. Synthesis

The three dopant-free HTMs were successfully synthesized as follows (Appendix A). The compound 3 and 6-bromo-4-hexyl-4*H*-thieno[3,2-*b*]indole were obtained according to previous reports with small modifications [38,39]. The detailed synthetic procedures and characterizations are given in the supporting information. The 6-bromo-4-hexyl-4*H*-thieno[3,2-*b*]indole was functionalized by Pd-catalyzed borylation and followed by Vilsmeier formylation reaction to afford compound 4. Then, π-bridges (compound 4) were introduced to the TAT core by Suzuki cross-coupling reaction. The final products of SGT-460, SGT-461, and SGT-462 were synthesized by condensing the aldehydes and the acceptors under Knoevenagel reaction conditions.

### 2.2. Photophysical and Electrochemical Properties 

The normalized UV-VIS absorption and photoluminescence (PL) spectra of the new compounds in tetrahydrofuran (THF) solution are shown in Figure 2a, and the corresponding parameters are listed in Table 1. Typically, for the dipole D-π-A structure molecules, charge-transfer absorption bands are found in the visible region, with the peak maximums centered at 478, 601, and 407 nm for SGT-460, SGT-461, and SGT-462, respectively. Photoluminescence spectra showed that all molecules have a large Stokes’ shift of around 100 nm, suggesting significant changes in the geometrical configuration of the molecules upon excitation. The optical bandgap (*E*_g_^opt^) values were estimated to be 2.45, 1.89, and 2.52 eV, for SGT-460, SGT-461, and SGT-462, respectively. The *E*_g_^opt^ variation can be ascribed to the combination of the changes in the accepting ability of the acceptor units (NM > ED > DI) with the conjugation length of the HTMs. The highest occupied molecular orbital (HOMO) energy levels of SGT-460, SGT-461, and SGT-462 were measured by cyclic voltammetry (CV) at −5.12, −5.32, and −5.41 eV, respectively (Figure 2b and Table 1). Also, the oxidation potential (*E*_ox_) variation can be attributed to the combination of the changes in the acceptor ability with the conjugation length of the HTMs. The HOMO energy levels are higher than a valence band of the photoactive perovskite layer MAPbCl_3−*x*_I*_x_* (−5.43 eV), guaranteeing efficient photogenerated charge transfer at the interface (Figure 2c) [21]. The HOMO level of SGT-462 was close to the valence band of the HOIP layer, which can result in the inefficient hole extraction from the HOIP to HTM layer [40]. The lowest occupied molecular orbital (LOMO) energy level values were determined to be −2.60, −3.43, and −3.03 eV for SGT-460, SGT-461, and SGT-462, respectively. These values were sufficiently lower than the conduction band of the MAPbCl_3−*x*_I*_x_* layer (−3.93 eV), ensuring the prevention of electron backflow from HOIP to the counter electrodes.

### 2.3. Thermal Analysis

Thermogravimetric analysis (TGA) and differential scanning calorimetry (DSC) measurements were performed (Appendix A and Appendix A) to evaluate the thermal properties of HTMs. The glass transition temperatures (T_g_) of SGT-460, SGT-461, SGT-462, and spiro-OMeTAD were 155, 158, 160, and 125 ℃, respectively. The glass transport temperature (T_g_) of all HTMs was higher than that of the spiro-OMeTAD. HTMs retaining a higher T_g_ can induce the lower tendency to oriented aggregation (crystallization) upon heating [16,41,42]. Also, the T_g_ of HTMs is an important factor for the thermal stability of amorphous films upon long-time device storage or heating. The degradation temperatures (T_d_) were found to be values of 345, 339, 357, and 406 ℃ for SGT-460, SGT-461, SGT-462, and spiro-OMeTAD, respectively. These results indicated that the HTMs were thermally stable up to ~330 ℃.

### 2.4. Hole Mobility Analysis

To determine the hole mobility of all HTMs, the space-charge limited current (SCLC) measurement was conducted with the fluorine doped tin oxide (FTO)/poly(3,4-ethylenedioxythiophene):polystyrene sulfonate (PEDOT:PSS)/HTM/Au device architecture (Appendix A and Table 1). The hole mobility values of SGT-460, SGT-461, and SGT-462 were determined to be 7.59 × 10^−5^ cm^2^ V^−1^ s^−1^, 5.13 × 10^−4^ cm^2^ V^−1^ s^−1^, and 7.61 × 10^−4^ cm^2^ V^−1^ s^−1^, respectively. Interestingly, the trend of hole mobility was comparable to the trend of the HOMO energy levels (SGT-460 < SGT-461 < SGT-462). In addition, these values were higher than the hole mobility value of spiro-OMeTAD without dopants (~10^−5^ cm^2^ V^−1^ s^−1^), indicating improved hole transporting properties through the more efficient π-π stacks [38]. The HOMO energy level and high hole mobility of these molecules make them good candidates for dopant-free HTMs in PrSCs.

### 2.5. Photovoltaic Properties

The three dopant-free HTMs were incorporated into conventional-type PrSCs with a stack of TiO_2_/MAPbCl_3-*x*_I*_x_* /HTM/Au to evaluate their photovoltaic properties (Figure 3a). The MAPbCl_3-*x*_I*_x_* perovskite layer was deposited according to a previous report (see Appendix A) [13]. The current density-voltage (*J-V*) characteristics of the PrSCs under air mass (AM) 1.5 G irradiation at 100 mW cm^−2^, and the incident photon-to-current conversion efficiency (IPCE) spectra are presented in Figure 3. The corresponding photovoltaic parameters are listed in Table 2. All dopant-free HTM-based PrSCs exhibited PCEs of 10.8%, 10.1%, and 11.7% for SGT-460, SGT-461, and SGT-462, respectively, while the PCEs of the spiro-OMeTAD-based devices were 6.8% without dopants (*J*_SC_ = 18.69 mA cm^−2^, *V*_OC_ = 0.929 V, *FF* = 39.1) and 17.8% with dopants (Appendix A and Appendix A, *J*_SC_ = 20.62 mA cm^−2^, *V*_OC_ = 1.094 V, *FF* = 75.5). This indicates that the end-capped acceptors worked as the dopant. In other words, the end-capped acceptors allowed self-doping HTMs by introduction of ionic character to them. The *V*_OC_, *J*_SC_, and *FF* values of the three dopant-free HTM-based PrSCs showed a similar tendency in the order of SGT-461 < SGT-460 < SGT-462 (*V*_OC_ of 0.898, 0.887, and 0.914 V; *J*_SC_ of 19.57, 19.15, and 20.01 mA cm^−2^; and *FF* of 61.5, 59.2, and 63.2, for SGT-460, SGT-461, and SGT-462, respectively). The histogram of three dopant-free HTM-based PrSCs is shown in Appendix A. It should be noted that he SGT-462-based PrSCs showed a higher *V*_OC_, *J*_SC,_ and *FF* values compared to those of SGT-460 and SGT-461, despite the insufficient driving force of 0.02 eV for regeneration from the HOMO energy level of SGT-462 to the HOIP. This phenomenon implies that the interfacial properties of three dopant-free HTMs mainly affected their photovoltaic performance, more than the intrinsic properties such as hole mobility and HOMO energy levels. Moreover, the lower *V*_OC_, *J*_SC,_ and *FF* values of the three dopant-free HTM-based devices, as compared to those of spiro-OMeTAD with dopants, could be due to a poor surface morphology of the HTM-coated film [24,43]. However, apparently the dopant-free HTM-based PrSCs showed significant enhancement in photovoltaic performance compared to the spiro-OMeTAD-based device without dopants. When adding the dopant, three SGT-HTM-based PrSCs showed low PCEs (Appendix A and Appendix A).

### 2.6. Surface Morphology Observation of HTM Films

The uniformity of the spin-coated HTM layers onto the perovskite film was affected by how well the HTMs dissolved in organic solvents and assembled on the HOIP layer, depending on molecular structure. It is well known that the film-forming ability of the HTMs is crucial to making good interfacial contact of MAPbCl_3-*x*_I*_x_*/HTM/Au, for which the nature of the interfacial contact between layers can significantly affect hole extraction and the charge recombination in devices. Therefore, the surface morphologies of the HTM films were observed by using field emission scanning electron microscopy (FE-SEM) and atomic force microscopy (AFM) to scrutinize the photovoltaic behavior of the SGT-460-, SGT-461-, and SGT-462-based PrSCs. Figure 4 displays the SEM images of the HOIP layer and spin-coated HTM films. Spiro-OMeTAD- and SGT-462-based films exhibited a good smooth and uniform layer on the MAPbCl_3-*x*_I*_x_* layer, while those of SGT-460 and SGT-461 showed a poor surface morphology (Figure 4). In particular, there were many pinholes on the SGT-461-based film. These results indicated the poor interfacial contact in the device, explaining the low *V*_OC_ and *FF* values of SGT-461-based devices.

The AFM results presented the same tendency as in the SEM observation (Figure 5). The largest surface roughness (RMS) of the bare HOIP layer was 14.8 nm. After the formation of HTM films on top of the HOIP layer, the RMS values of the HTM-coated films were significantly decreased with the order of spiro-OMeTAD (2.3 nm) < SGT-462 (2.8 nm) < SGT-460 (5.4 nm) < SGT-461 (6.7 nm). To elucidate the variation in surface morphologies, the solubility test of three dopant-free HTMs was conducted with three different solvents (Appendix A). Three dopant-free HTMs were almost insoluble in toluene and 1,2-dichlorobenzene and less soluble in 1,1,2,2-tetrachloroethane. The solubility of SGT-460, SGT-461, and SGT-462 in 1,1,2,2-tetrachloroethane were 16.2, 9.8, and 20.7 mg mL^−^^1^, respectively, exhibiting the same trend in RMS values. As a result, it was concluded that the planar structure of the ED and DI moieties affected the solubility of molecules and their surface morphology, thereby resulting in the poor interfacial contact in the PrSC device with low *V*_OC_ and *FF* values [44].

### 2.7. Time-Resolved Photoluminescence Measurements

The time-resolved photoluminescence (TR-PL) measurements were carried out to study the interfacial electron transfer process between the three dopant-free HTMs and HOIP layer [21]. The TR-PL spectra are displayed in Figure 6 and the fitted and calculated parameters are listed in Appendix A. The PL lifetimes of the pristine HOIP films, HOIP/spiro-OMeTAD, HOIP/SGT-460, HOIP/SGT-461, and HOIP/SGT-462 were about 49.2, 11.27, 16.39, 19.94, and 15.17 ns, respectively. The shorter decay time of the SGT-462-based device compared to those of the SGT-460- and SGT-461-based devices, demonstrating that SGT-462 had a more effective hole extraction ability and less charge recombination [45]. The charge-transfer efficiencies (CTE) were calculated to be 77.1, 66.7, 59.5, and 69.2% for spiro-OMeTAD-, SGT-460-, SGT-461-, and SGT-462-based devices, respectively, which correspond to the trend in PCE (spiro-OMeTAD > SGT-462 > SGT-460 > SGT-461). Moreover, these trends in CTE values of three dopant-free SGT HTMs can explain their trends in *J*_SC_ values. The SGT-461-based film exhibited good hole mobility, but their PrSCs showed lower *J*_SC_ values than those of other SGT HTMs. This indicates that the interfacial properties were the main factor that affected the *J*_SC_ values with a minor influence from the hole mobility. Interestingly, the SGT-462-based device showed a lower decay time than that of the spiro-OMeTAD, despite its good morphology, unlike those of SGT-460 and SGT-461. This phenomenon may be explained by the low driving force between the HOMO energy level of SGT-462 and the valence band of the MAPbCl_3-*x*_I*_x_*. 

## 3. Conclusions

In summary, new dopant-free D-π-A structural hole-transporting materials (HTMs), named as SGT-460, SGT-461, and SGT-462, incorporating a planar type triazatruxene (TAT) core, thieno[3,2-*b*]indole (TI) π-bridge and three different acceptors, 3-ethylthiazolidine-2,4-dione (ED), 3-(dicyano methylidene)indan-1-one (DI), and malononitrile (MN), were designed and synthesized for application in PrSCs. Their photophysical, electrochemical, and photovoltaic behaviors were studied and compared with the reference HTM of spiro-OMeTAD. The HOMO energy levels of the three HTMs decreased in the order of SGT-460 (5.12 eV) < SGT-461 (5.32 eV) < SGT-462 (5.41 eV) along with the variation of the acceptor ability of HTM molecules. Their HOMO energy levels were positioned for efficient hole extraction from a MAPbCl_3−_*_x_*I*_x_* layer (5.43 eV). The hole mobility values of the HTMs without dopants were determined to be 7.59 × 10^−5^ cm^2^ V^−1^ s^−1^, 5.13 × 10^−4^ cm^2^ V^−1^ s^−1^, and 7.61 × 10^−4^ cm^2^ V^−1^ s^−1^ for SGT-460-, SGT-461-, and SGT-462-based films. In addition, the glass transition temperature (T_g_) of all HTMs showed higher than that of the spiro-OMeTAD. To clarify the variation in *V*_OC_ and *FF* values, the surface morphology of the HTMs film on the top of MAPbCl_3-_*_x_*I*_x_* layer were observed by SEM and AFM. Spiro-OMeTAD- and SGT-462-based films exhibited a smooth surface and low RMS values, respectively, while the SGT-460- and SGT-461-based films exhibited a rough surface and high RMS values. The PrSCs based on SGT-462 exhibited almost 2 times higher efficiency than that of dopant-free spiro-OMeTAD-based device, due to the high intrinsic hole mobility and excellent film-forming ability of SGT-462. These results could explain the reason for the low *V*_OC_ and *FF* values of SGT-460 and SGT-461. Moreover, low solubility of SGT-460 and SGT-461 in 1,1,2,2-tetrachloroethane explained their poor surface morphology. Finally, the PL decay time of HTMs were measured by TR-PL measurements. The trend charge-transfer efficiencies (CTE) of HTMs were in good agreement with the trends in *V*_OC_ and *FF*, indicating that the interfacial properties of dopant-free HTMs mainly affected the *J*_SC_. This study of structural engineering of TAT-based, dopant-free HTMs with different acceptor moieties provides insights for developing efficient dopant-free, small molecule HTMs in the future.

## Figures and Tables

**Figure 1 nanomaterials-10-00936-f001:**
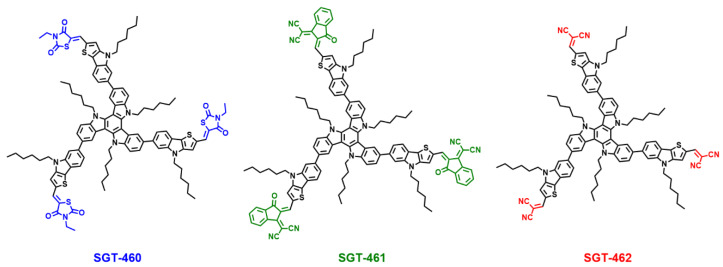
Molecular structures of three dopant-free hole-transporting materials (HTMs).

**Figure 2 nanomaterials-10-00936-f002:**
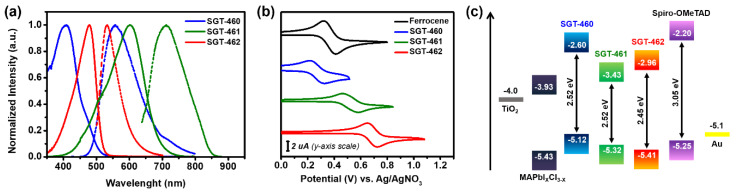
(**a**) UV-VIS absorption (solid line) and emission (dashed line) spectra of SGT-460, SGT-461, and SGT-462 measured in THF solution. (**b**) Cyclic voltammograms of ferrocene (Fc, an external reference), SGT-460, SGT-461, and SGT-462. (**c**) Energy diagram with respect to components used in PrSC devices.

**Figure 3 nanomaterials-10-00936-f003:**
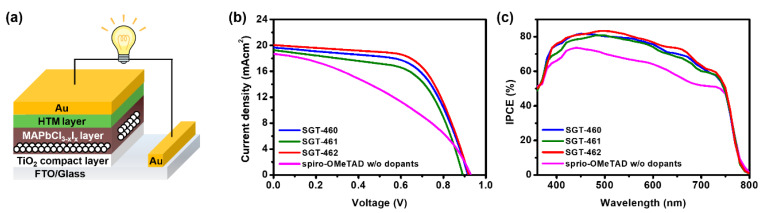
(**a**) Schematic device architecture for PrSC. (**b**) *J-V* curves of the PrSCs using SGT-460, SGT-461, SGT-462, and spiro-OMeTAD (with dopants/without dopants) as HTMs. (**c**) The corresponding incident-photon-to-electron conversion efficiency (IPCE) spectra.

**Figure 4 nanomaterials-10-00936-f004:**
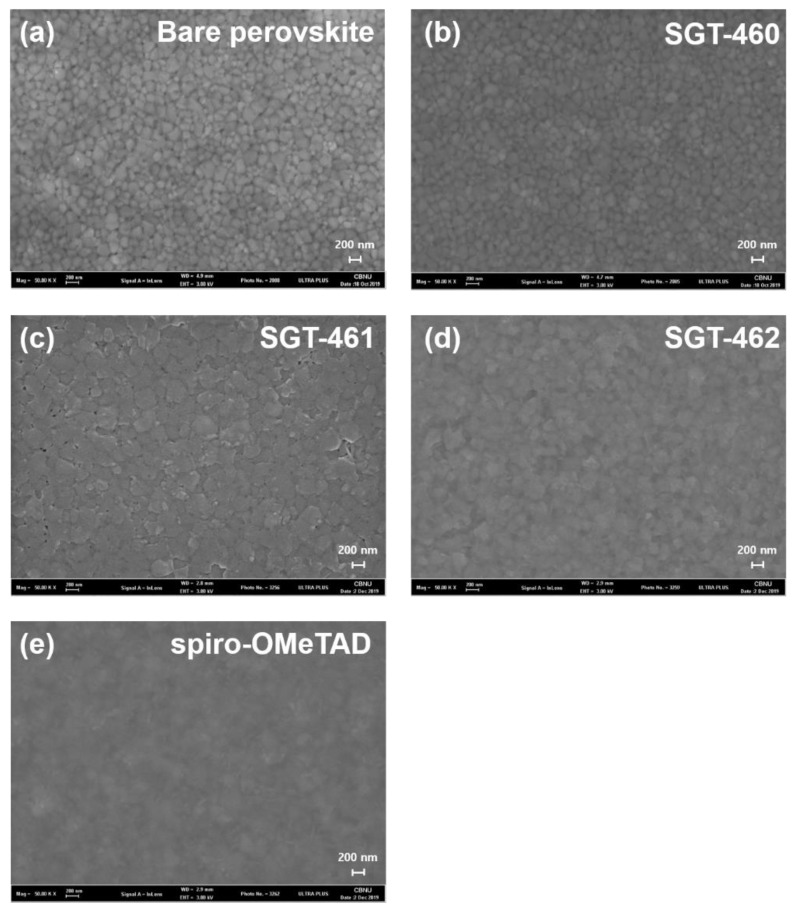
(**a**) Top surface SEM images of the methylammonium (MA) lead mixed halide perovskite (MAPbCl_3−*x*_I*_x_*) film. (**b**–**e**) The top view of various HTMs’ layers on the perovskite films, (**b**) SGT-460, (**c**) SGT-461, (**d**) SGT-462, and (**e**) spiro-OMeTAD.

**Figure 5 nanomaterials-10-00936-f005:**
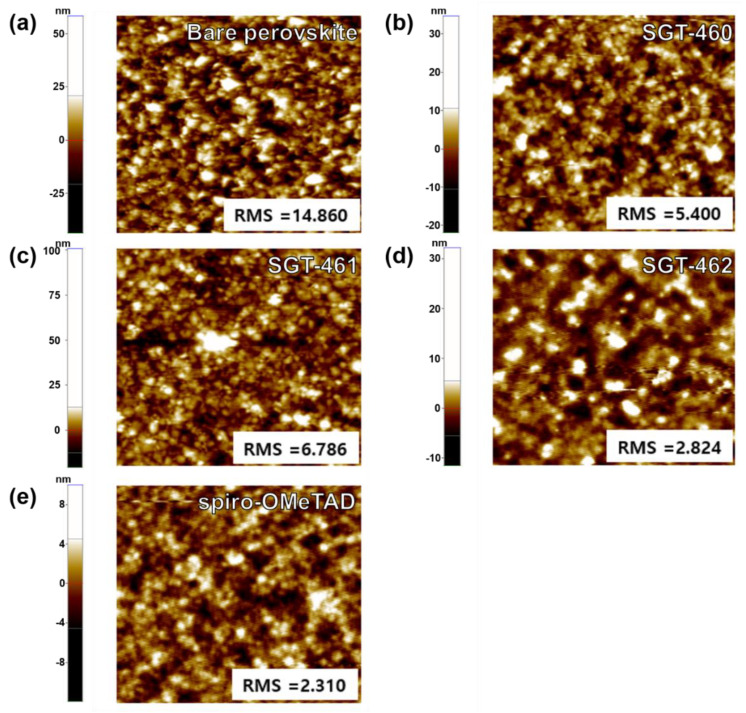
Atomic Force Microscope (AFM) images of the top morphology of various films. (**a**) MAPbCl_3-*x*_I*_x_* film. (**b**–**e**) The top morphology of various HTM layers on the perovskite films: (**b**) SGT-460, (**c**) SGT-461, (**d**) SGT-462, and (**e**) spiro-OMeTAD.

**Figure 6 nanomaterials-10-00936-f006:**
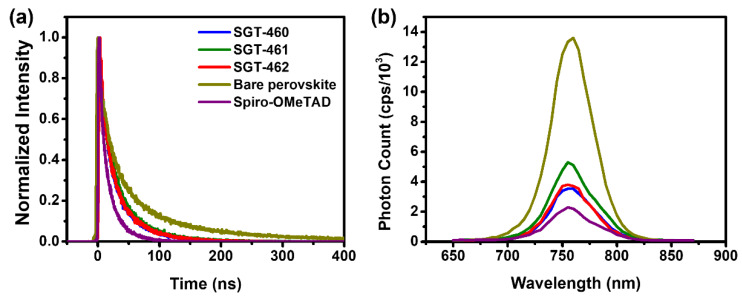
(**a**) Time-resolved photoluminescence (PL) spectra of pristine HOIP and HOIP/HTM bilayer, with excitation at 493 nm and monitoring at 755 nm. (**b**) The steady-state PL spectra of corresponding devices, with excitation at 600 nm.

**Table 1 nanomaterials-10-00936-t001:** Photophysical and electrochemical data of SGT dopant-free HTMs and 2,2′,7,7′-tetrakis[*N*,*N*-di(4-methoxyphenyl)amino]-9,9′-spirobifluorene (spiro-OMeTAD).

HTM	*λ*_abs max_ (nm)	*ε* (M^−1^ cm^−1^)	*λ*_em max_ (nm)	*E*_g_^opt 1^ (eV)	*E*_ox_^2^ (V vs. NHE)	*E*_HOMO_^3^ (eV)	*E*_LUMO_^4^ (eV)	Hole Mobility ^5^ (cm^2^ V^−1^ s^−1^)
SGT-460	407	25390	557	2.52	0.512	−5.12	−2.60	7.59 × 10^−5^
SGT-461	607	13244	713	1.89	0.816	−5.32	−3.43	5.13 × 10^−4^
SGT-462	478	17424	532	2.45	0.910	−5.41	−3.03	7.61 × 10^−4^
Spiro-OMeTAD ^6^	387	60530	418	3.05	0.751	−5.25	−2.20	7.59 × 10^−5^

^1^ Estimated from the cross point of the normalized absorption and emission spectra measured in THF, by *E*_g_^opt^ = 1240/*λ*_0−0_. ^2^ Potentials were referenced to an external Fc/Fc^+^ standard (0.63 V *vs.* NHE). ^3^ Calculated by *E*_HOMO_ (eV) = −4.5 − *E*_ox_. ^4^
*E*_LUMO_ was estimated by *E*_HOMO_ + *E*_g_^opt^. ^5^ Measured in the space-change limited current (SCLC) regime and fitted using the Mott–Gurney law. ^6^ According to [14].

**Table 2 nanomaterials-10-00936-t002:** Photovoltaic performances of the PrSCs based on dopant free SGT-460, SGT-461, SGT-462, and spiro-OMeTAD.

HTM	*J*_SC_ (mA cm^−2^)	*V*_OC_ (V)	*FF* (%)	PCE (%) ^1^
SGT-460	19.57 ± 0.11	0.898 ± 3.02	61.5 ± 0.22	10.8 ± 0.07
SGT-461	19.15 ± 0.12	0.887 ± 3.74	59.2 ± 0.27	10.1 ± 0.10
SGT-462	20.01 ± 0.11	0.914 ± 4.83	63.2 ± 0.35	11.7 ± 0.06
spiro-OMeTAD w/o dopants	18.69 ± 0.14	0.929 ± 1.43	39.1 ± 0.1	6.8 ± 0.08

^1^ The averaged PCEs were calculated using six different cells and measured under simulated AM 1.5 G irradiation.

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
