# Peer review of "Dopant-Free Triazatruxene-Based Hole Transporting Materials with Three Different End-Capped Acceptor Units for Perovskite Solar Cells"

_nanomaterials, 2020, doi:10.3390/nano10050936_

Round 1

Reviewer 1 Report

In this manuscript, the authors reported three dopant-free HTMs with various end-capped acceptors for perovskite solar cells. The effects of end-capped acceptors in D-π-A structured HTMs on molecular and photovoltaic properties were investigated via photophysical and electrochemical experiments. The cells based on three HTMs are shown to have moderate efficiency. This achievement is worthy and the results will be helpful for the related researchers in the field. However, in order to publish this work, several important points have to be addressed:

  1. In the introduction, part authors should appropriately cite the latest literature based on dopant-free HTMs, i.e. Nano Energy 72, 2020, 104708; Adv. Electron. Mater. 2020, 1900884; ACS Appl. Energy Mater. 2019, 2, 10, 7070-7082; Adv. Mater. 2017, 29, 1606555; Chem. Sci., 2019,10, 6748-6769.
  2. Authors should carefully read the manuscript and correct all the typos, there are plenty along with the text.
  3. The statement "...end-capped acceptors work as the dopant" is ambiguous, the authors should clarify and explain it in detail.
  4. The part of the title "...stable and efficient perovskite solar cells" is misleading since there is no single sentence in the manuscript regarding the stability of the devices prepared with the new HTMs. In order to say that authors should provide stability test data under full sunlight for at least 1000h.

Author Response

Reviewer 1

In this manuscript, the authors reported three dopant-free HTMs with various end-capped acceptors for perovskite solar cells. The effects of end-capped acceptors in D-π-A structured HTMs on molecular and photovoltaic properties were investigated via photophysical and electrochemical experiments. The cells based on three HTMs are shown to have moderate efficiency. This achievement is worthy and the results will be helpful for the related researchers in the field. However, in order to publish this work, several important points have to be addressed:

Many thanks for very thoughtful comments and detailed comments below.

  1. In the introduction, part authors should appropriately cite the latest literature based on dopant-free HTMs, i.e. Nano Energy 72, 2020, 104708; Adv. Electron. Mater. 2020, 1900884; ACS Appl. Energy Mater. 2019, 2, 10, 7070-7082; Adv. Mater. 2017, 29, 1606555; Chem. Sci., 2019,10, 6748-6769.

Answer: Thank you for your helpful comment. We added the references to the introduction section.

  1. Authors should carefully read the manuscript and correct all the typos, there are plenty along with the text.

Answer: Thank you for your detailed comment. We carefully read the manuscript and corrected all the typos.

  1. The statement "...end-capped acceptors work as the dopant" is ambiguous, the authors should clarify and explain it in detail.

Answer: Thank you for your detail comment. The sentence "...end-capped acceptors work as the dopant" just described the working of the dopant-free HTM based PrSCs without dopants. We added the sentence as follows:

“In the other word, the end-capped acceptors allow “self-doping HTMs by introduction of ionic character to them.”  

  1. The part of the title "...stable and efficient perovskite solar cells" is misleading since there is no single sentence in the manuscript regarding the stability of the devices prepared with the new HTMs. In order to say that authors should provide stability test data under full sunlight for at least 1000h.

Answer: Thank you for your thoughtful comment. We deleted the part of the title “stable and efficient”

Reviewer 2 Report

Kil et al. report on three new dopant-free D-π-A hole transporting materials, namely SGT-460, SGT-461 and SGT-462. The compounds were obtained through a facile synthetic route by crosslinking the electron-donating triazatruxene with the thieno[3,2-b]indole spacer, and finally linked to different electron-acceptor units. The new derivatives were used, without dopants, in perovskites solar cells (MAPbIxCl3-x) and showed moderate performances. This manuscript is very interesting, the contribution is valuable at the apply level. The manuscript is clearly written and provides manifold data supporting the conclusions. I am happy to recommend publication after the following minor revision:

  1. A more recent review should be included as a reference: Soc. Rev., 2018, 47, 8541; Chem. Sci., 2019,10, 6748-6769
  2. The characterization of the new HTM is very poor, please provide a complete characterization of the new derivatives not only 1H and Mass spectrometry.
  3. The authors should include a statistic analysis. How many devices were made for each HTM?
  4. The concentration employed for each HTM should be included.
  5. Did the author use these HTMs with dopants? If so, please include it in the manuscript.

Author Response

Reviewer 2

Kil et al. report on three new dopant-free D-π-A hole transporting materials, namely SGT-460, SGT-461 and SGT-462. The compounds were obtained through a facile synthetic route by crosslinking the electron-donating triazatruxene with the thieno[3,2-b]indole spacer, and finally linked to different electron-acceptor units. The new derivatives were used, without dopants, in perovskites solar cells (MAPbIxCl3-x) and showed moderate performances. This manuscript is very interesting, the contribution is valuable at the apply level. The manuscript is clearly written and provides manifold data supporting the conclusions. I am happy to recommend publication after the following minor revision:

 Many thanks for very helpful comments below.

  1. A more recent review should be included as a reference: Soc. Rev., 2018, 47, 8541; Chem. Sci., 2019,10, 6748-6769

Answer: Thank you for your helpful comment. We added two references to the revised manuscript.

  1. The characterization of the new HTM is very poor, please provide a complete characterization of the new derivatives not only 1H and Mass spectrometry.

Answer: Thank you for your thoughtful comment. We have missed obtaining13C NMR spectra for the new derivatives. At this moment, all new derivatives have been used up for device fabrication, cell optimization and cell performance measurements so that we could not add 13C NMR spectra. I am really sorry about it.

  1. The authors should Include a statistic analysis. How many devices were made for each HTM?

Answer: Many thanks for your detail comment. We optimized the dopant-free HTMs based PrSCs with various factors, such as solvent spin coating rpm. We uploaded a statistic analysis (Figure S5) using six different cells to the supporting information and added “The histogram of three dopant-free HTM based PrSCs are showed in Figure S5.” in the revised manuscript-SI.

  1. The concentration employed for each HTM should be included.

Answer: Many thanks for your detail comment. We added a sentence to ‘Fabrication of perovskite solar cells section’ in supporting information as follow:

The concentration of three HTMs were 16.2, 9.8 and 20.7 mg in 1ml of 1,1,2,2-tetrachloroethane for SGT-460, SGT-461 and SGT-462, respectively. The concentration of spiro-OMeTAD was 72.3 mg/1 mL chlorobenzene. When using the dopants, 17.5 µL of tris(bis(trifluoro-methyl-sulfonyl)imide) (Li-TFSI) stock solution (520 mg/1 mL in acetonitrile), 28.8 µL of tert-butylpyridine (t-BP) were added to HTMs solution as additives.

  1. Did the author use these HTMs with dopants? If so, please include it in the manuscript.

Answer: Many thanks for your detail comment. We already made the dopant-free HTMs based PrSCs with dopant. However, we could not strongly address it due to their low efficiency as follows:

We added Figure S4 to supporting information and their corresponding parameters were added to Table S2, and the sentence “When adding the dopant, three SGT-HTMs based PrSCs showed low PCEs (Figure S4).” were added in the revised manuscript.

Reviewer 3 Report

In this contribution the authors report a quite comprehensive study of the photophysical, electrochemical, and photovoltaic properties of a new family of donor-acceptor organic materials proposed as hole-transporting materials (HTMs) for perovskite-based solar cells (PSCs). The paper is well structured and written and the conclusions are well supported by the results presented. I found the manuscript of wide interest for the scientific community working on PSCs. The subject of the work is in line with the scope of the journal, and the paper is suitable for publication in Nanomaterials after addressing the following concerns.

1) The main concern corresponds to the motivation of the paper. The authors motivate the interest of the new systems as dopant-free HTMs with PCEs higher that the archetype HTM spiro-OMeTAD. However, there are many HTM systems in the bibliography that when used without any doping in PSCs exhibit significantly higher PCEs than those reported here. The authors must revise in the introduction the different strategies used to design dopant-free HTMs and should compare their results with those previously reported

2) It is really strange that no section about the synthesis of the new HTMs and no reference to that synthesis appears in the manuscript.

3) The estimation of the HOMO and LUMO energies in Table 1 from oxidation potentials and optical data is wrongly described. The HOMO energies included in the Table cannot be obtained from the expression given, and the E0,0 values needed to calculate the energy of the LUMO are not quoted.

4) The calculations do not really enrich the discussion and they constitute only accessory data. They are not discussed in connection to the experimental results. The approach used is not explained and the reference given (14) has no relation to the method used. I suggest to completely remove the calculations.

Author Response

Reviewer 3

In this contribution the authors report a quite comprehensive study of the photophysical, electrochemical, and photovoltaic properties of a new family of donor-acceptor organic materials proposed as hole-transporting materials (HTMs) for perovskite-based solar cells (PSCs). The paper is well structured and written and the conclusions are well supported by the results presented. I found the manuscript of wide interest for the scientific community working on PSCs. The subject of the work is in line with the scope of the journal, and the paper is suitable for publication in Nanomaterials after addressing the following concerns.

Thank you for very helpful comments below.

  1. The main concern corresponds to the motivation of the paper. The authors motivate the interest of the new systems as dopant-free HTMs with PCEs higher that the archetype HTM spiro-OMeTAD. However, there are many HTM systems in the bibliography that when used without any doping in PSCs exhibit significantly higher PCEs than those reported here. The authors must revise in the introduction the different strategies used to design dopant-free HTMs and should compare their results with those previously reported

Answer: Many thanks for your helpful comment. We revised the introduction section as follows:

“Most of efficient dopant-free HTMs have the donor−π−acceptor (D-π-A) architecture because they allow the low band gap, ionic character enhancement by avoiding the use of dopants and strong dipolar intermolecular interaction. Based on the D-π-A architecture, many researchers approach with various strategies to solve the above disadvantages. One of the strategies is the modification of molecular geometry, which can affect its dipole moment, solubility and film quality. The linear, star-shaped, and butterfly-shaped geometry are representative. The other is the modification of moieties such as core unit, electron donor, π-bridge and acceptor. The dopant-free HTMs based on various moieties such as diphenylamine, triphenylamine, carbazole as electron donors, thiophene, 3,4-ethylenedioxythiophene, naphtho-dithiophene, spiro[fluorene-9,9′-xanthene] and triazatruxene (TAT) as core units, electron donors and π-bridges, diketopyrrolopyrrole, isoindigo and benzothiadiazole as electron acceptors, have been applied in PrSCs” in introduction part.

However, we can’t accept description about the comparison between our result and those of previously reports because the characterization and device fabrication conditions are pretty different. 

  1. It is really strange that no section about the synthesis of the new HTMs and no reference to that synthesis appears in the manuscript.

Answer: Many thanks for your comment. We added the synthesis section in manuscript as follow:

“The three dopant-free HTMs were successfully synthesized (Scheme S1). The TAT core (Compound 3) and 6-bromo-4-hexyl-4H-thieno[3,2-b]indole were obtained according to previous reports with small modifications(38, 39). The detailed synthetic procedures and characterizations are given in the supporting information. 6-bromo-4-hexyl-4H-thieno[3,2-b]indole was functionalization by Pd catalyzed borylation and by Vilsmeier formylation reaction to afford compound 4. Then, π-bridges (compound 4) were introduced to the TAT core by Suzuki cross-coupling reaction. The final products SGT-460, SGT-461 and SGT-461 were synthesized by condensing the aldehydes and the acceptors under Knoevenagel reaction conditions.”

  1. The estimation of the HOMO and LUMO energies in Table 1 from oxidation potentials and optical data is wrongly described. The HOMO energies included in the Table cannot be obtained from the expression given, and the E0,0 values needed to calculate the energy of the LUMO are not quoted.

Answer: Thank you for your detailed comment. We found the errors that you point out. Thus, we carefully determined the HOMO and LUMO energy values from original data, again. We revised Table 1 in the revised manuscript.

  1. The calculations do not really enrich the discussion and they constitute only accessory data. They are not discussed in connection to the experimental results. The approach used is not explained and the reference given (14) has no relation to the method used. I suggest to completely remove the calculations.

Answer: Many thanks for very thoughtful comments. We deleted the theoretical approaches section in the revised manuscript.

Round 2

Reviewer 1 Report

The authors carefully fixed all the points that have been raised and improved the quality of the manuscript, therefore I suggest accepting this work in the present form.

Reviewer 2 Report

The manuscript can be accepted.

Reviewer 3 Report

The authors have adequately addressed the points raised in my previous review and I therefore consider the paper suitable to be published in Nanomaterials as it is.